# A high-fidelity quantum matter-link between ion-trap microchip modules

M. Akhtar[1,2,5], F. Bonus [2,3,5], F. R. Lebrun-Gallagher [1,2,5], N. I. Johnson [1], M. Siegele-Brown[1], S. Hong[1], S. J. Hile [1], S. A. Kulmiya[1,4], S. Weidt[1,2] & W. K. Hensinger [1,2] ✉

System scalability is fundamental for large-scale quantum computers (QCs) and is being pursued over a variety of hardware platforms. For QCs based on trapped ions, architectures such as the quantum charge-coupled device (QCCD) are used to scale the number of qubits on a single device. However, the number of ions that can be hosted on a single quantum computing module is limited by the size of the chip being used. Therefore, a modular approach is of critical importance and requires quantum connections between individual modules. Here, we present the demonstration of a quantum matter-link in which ion qubits are transferred between adjacent QC modules. Ion transport between adjacent modules is realised at a rate of 2424 s$^{-1}$ and with an infidelity associated with ion loss during transport below $7 \times 10^{-8}$. Furthermore, we show that the link does not measurably impact the phase coherence of the qubit. The quantum matter-link constitutes a practical mechanism for the interconnection of QCCD devices. Our work will facilitate the implementation of modular QCs capable of fault-tolerant utility-scale quantum computation.

Platforms using trapped atomic ions form an exceptional foundation on which QCs and quantum simulators can be developed[1]. Encoding qubits in the internal electronic states of trapped ions offers the highest quantum gate fidelities and the longest coherence times when compared to other physical implementations[2–8].

So far, small-scale trapped-ion QCs with up to 10s of qubits have been demonstrated[9–12]. At this scale, quantum logic can be realised using a single linear ion crystal as a qubit register. Multi-qubit operations are mediated by the Coulomb interaction within the crystal. However, as the crystal size increases, limitations on the motional mode density make scaling a single register to larger qubit numbers challenging[13]. One architecture that allows for multiple qubit registers in its design is the quantum charge-coupled device (QCCD) which consists of an array of segmented electrodes[14,15]. Locations or zones within a single device can be allocated specific functions, such as quantum information processing, memory and read-out. In this configuration, a shared RF pseudopotential provided by a single pair of RF electrodes delivers the necessary radial confinement, while confinement in the axial direction is created by quasistatic potentials applied to the remaining segmented electrodes. This allows for small qubit registers to be interfaced to one another via mobile ions. For quantum information processing using a shuttling-based trapped ion architecture, there are two sources of infidelity to consider. The first source of infidelity is the decay of the quantum state during the ion transport operation. Work by Kaufmann et al. has already demonstrated that ion transport operations within a single device can give rise to a state fidelity of 99.9994($^{+6}_{-7}$)% per transport operation[16]. The second possible source of infidelity is the infidelity associated with ion loss during transport.

Furthermore, the QCCD architecture has been used to implement high-fidelity quantum logic, and can be paired with laser-free gate schemes[4,11,17]. Recently, QCCD architectures have been used to demonstrate key steps towards the fault-tolerant operation of a trapped ion quantum computer[11,18,19] and stand-alone near-term devices

[1]Sussex Centre for Quantum Technologies, University of Sussex, Brighton BN1 9QH, UK. [2]Universal Quantum Ltd, Brighton BN1 6SB, UK. [3]Department of Physics and Astronomy, University College London, London WC1E 6BT, UK. [4]Quantum Engineering Centre for Doctoral Training, University of Bristol, Bristol BS8 1TH, UK. [5]These authors contributed equally: M. Akhtar, F. Bonus, F. R. Lebrun-Gallagher. ✉e-mail: w.k.hensinger@sussex.ac.uk

offer an attractive platform for the execution of restricted noisy intermediate-scale quantum algorithms[20,21] and simulations[22].

To unlock many of the anticipated applications of a QC within the necessary level of error correction, far larger qubit numbers will be required than are available on current devices[23–25]. For example, simulations of the FeMoco molecule could lead to a better understanding of nitrogen fixation for the production of ammonia in fertilisers, but simulating its ground state would require more than $10^6$ qubits[24]. However, incorporating these large numbers of qubits into a single QCCD does not appear feasible given the size limitations of a single device. For instance, an ion trap array occupying an area of $90 \times 90$ mm$^2$ could be fabricated on a standard 150-mm silicon wafer[17]. A 1296 X-junction array could then be realised using X-junction trap structures with a footprint of $2.5 \times 2.5$ mm$^2$ per X-junction. Considering a practical upper bound where up to 20 trapped ion qubits are controlled per junction, ~$2.6 \times 10^4$ qubits could be housed on a single device. While a larger number of qubits could be incorporated onto wafers with sizes up to 450 mm, the fabrication of these devices is increasingly difficult to implement with increasing wafer size. A realistic ion-trap QC architecture must therefore be constructed from a network of QC modules and offer inter-module connection rates that are orders of magnitude faster than qubit decoherence times.

Thus far, the only experimentally demonstrated method used to connect trapped-ion QC modules relies on photonic links[26–29]. This optical interface permits the heralded, probabilistic distribution of entanglement between remote modules. Figure 1a briefly describes the processes involved in implementing a photonic link for an ion-trap QC. Photonic interconnects between two QC modules have been realised with an entanglement connection rate of 182 s$^{-1}$ at an entanglement fidelity of 94%[27].

Large optical switch arrays and wavelength conversion schemes offer a pathway towards connecting multiple modules together for large scale QCs. However, implementing these solutions in a fault-tolerant QC, would currently achieve an effective connection rate that is ~2 orders of magnitude less than the raw entanglement rate (see Methods). This reduces the rate to a level that might be too slow for implementation in a practical quantum computer (~1 s). While techniques to improve photon collection efficiencies such as on-chip cavities could

increase the entanglement rate, their integration within ion-trap modules remains an unsolved and difficult challenge[30]. By contrast, an alternative implementation relying on the entanglement of trapped ion chains through heralded single-photon transfer in shared overlapped cavities offers fast ($\mu$s) interaction rates[31]. Nevertheless, the approach is likely to remain constrained to tens of thousands of qubits after which slower photonic links may be relied upon for further expansion.

Lekitsch et al. proposed an alternative approach to scaling trapped-ion QCs where ion qubit transport between independent modules is mediated by electric fields[17]. Figure 1b highlights the processes involved in using a quantum matter-link for an ion-trap QC. In the illustration, surface-electrode ion-trap modules are depicted with an electrode structure that spans to the module's edge. When the electrodes on the edge of each module are aligned with respect to its neighbour, ions can be transported with translating potentials from one module, across the inter-module gap, to the next[17]. Figure 2 shows an example of how this method might be implemented on a large scale. It illustrates a conceptual modular QCCD architecture within which quantum information is not only distributed between zones within a single module, but also between modules, where radial confinement is provided by discontinuous RF electrode pairs. Using this architecture, it is then possible to build a quantum network of tessellated QC modules. One of the first steps towards realising the deterministic transport of individual trapped ions between RF ion-trap modules was made by Stopp et al., where an ion was ejected and recaptured by the same module, demonstrating a recapture fidelity of 95.1%[32].

In this article, we demonstrate the transfer of trapped ions between two quantum computing modules at a transfer rate of 2424 s$^{-1}$ over a total distance of 684 $\mu$m, and with an infidelity associated with ion loss during transport of less than $7 \times 10^{-8}$. Furthermore, we demonstrate that there is no measurable loss of coherence of the qubit associated with the transport operation, therefore realising a high-fidelity coherent quantum matter-link between adjacent quantum computer modules.

## Results
### Experimental set-up
We address the challenge of connecting independent QC modules by using two linear surface-electrode Paul traps. Both ion-trap microchip

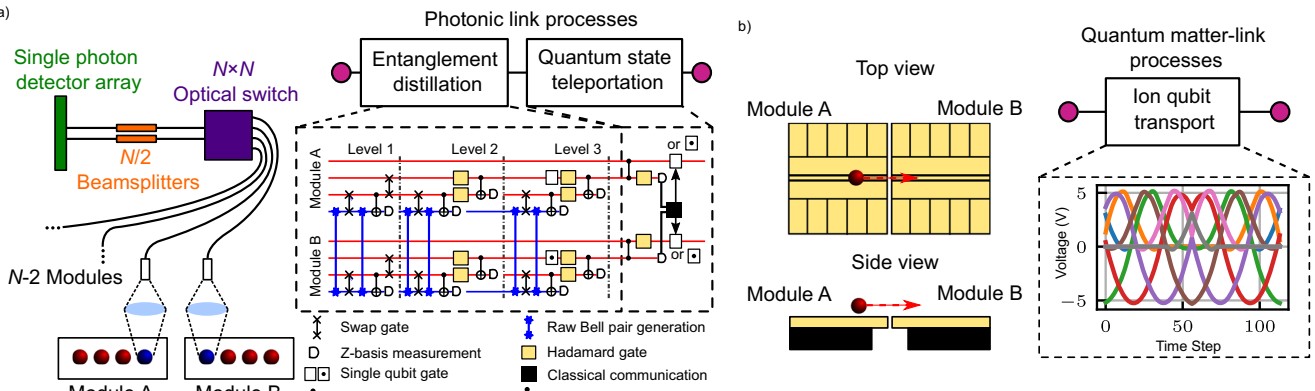

**Fig. 1 | An illustration of the process for connecting two modules (Module A and Module B) for the purposes of a modular ion-trap QC. a** Photonic links use probabilistic, heralded entanglement which is generated from the interference of emitted photons from each module. The red and blue spheres denote different ion species. An $N \times N$ optical switch links the modules, where $N$ is the number of modules. The emitted photons interfere at the beam splitter and a single photon detector array is used to herald the generation of entanglement between modules. In order to generate a high-fidelity link between modules a distillation process is required. The dashed box represents a proposal for a 3-level distillation process that would take a 94% entanglement fidelity to 99.7% fidelity, more details can be found in Ref. [48]. After distillation, quantum teleportation can then

be used to map the qubit state between modules, thereby completing the information transfer. A similar protocol involving measurement, single qubit rotations and classical communications can also be used to execute a remote two-qubit gate. **b** Linking modules using ion qubit transport. DC voltage waveforms control the ion motion such that the ion qubit is physically transported between the modules. The dashed box contains a plot of the voltage shuttling waveforms used in this work, each plot colour denotes the voltage evolution of a different DC electrode. Here the modules are depicted as surface traps but other geometries are also applicable. This method does not require quantum gates. Furthermore, for information transfer in a QC architecture based on quantum matter-links, multi-species gates are not necessary.

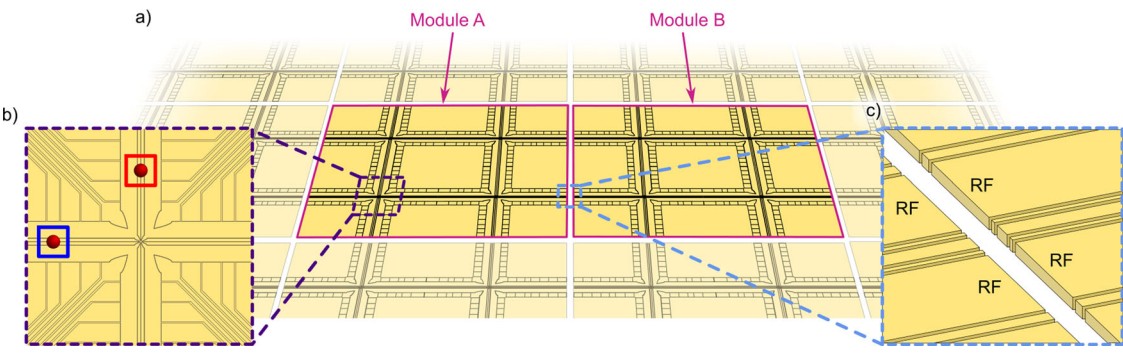

**Fig. 2 | An illustration of a small section of a modular version of the QCCD quantum computer architecture. a)** Two modules ("Module A" and "Module B") are shown as fully opaque and parts of ten other modules are shown as partially transparent. As an example, each module contains 4 X-junctions and is structured such that it tessellates with neighbouring modules. A realistic implementation may feature many more X-junctions per module. Inset **b)** shows a single X-junction on a module. For this architecture, specific areas of the X-junction are associated with certain functions and are connected by mobile ions. For instance, a qubit may be located in a gate zone for quantum logic operations (blue box) while another qubit may be stored in, or moved to, a memory zone (red box). Inset **c)** depicts the gap separating the modules. Distinct RF electrode pairs extend out to the edge of the modules to create a radially confining potential such that ions can be transported to the neighbouring module.

modules have been fabricated on a silicon substrate and feature electrode structures that extend to the edge of the inter-module gap (see Methods) while the substrate recedes ~75 μm away from the gap to facilitate module alignment. When the modules are aligned, this electrode configuration allows for the confining potential to extend over the inter-module gap, creating an electric field interface between the two modules.

One of the ion-trap modules ('Alice', left in Fig. 3) is rigidly mounted to the vacuum chamber via a heatsink circulating cryogenic helium gas such that the trap operates at 36–42 K[33]. The second ion-trap module ('Bob', right in Fig. 3) is cooled via a flexible copper braid forming a thermal link between the two modules. Bob is mounted to an in-vacuum three-axis piezo stage assembly (Physik Instrumente Ltd, P625.1 and P625.2). When considering many ion trap QCCD modules tessellated into a large-scale architecture as presented in Fig. 2, small UHV- and cryogenic-compatible XYZ piezo actuators will be required to be installed beneath each module[17]. These may be engineered from a combination of compact shear piezo actuators, providing the sufficient travel range to compensate for module drift that can arise from temperature changes while maintaining high module alignment accuracy.

In this work, the selection of piezo actuators was not constrained by the size of the module as only two modules required alignment. Instead, the implemented piezo stage assembly was chosen for its fine 5-nm positioning accuracy and large travel range of 600 μm which provides additional experimental flexibility. However, since the measurement of the alignment of the two modules is done optically, the precision of the module alignment in the x−y plane is limited by the imaging system. To image the modules a lens system with × 13 magnification is used in conjunction with an sCMOS camera. The imaging system has a spatial resolution of 0.5 μm which leads to an alignment error of 1 μm in the x−y plane. In the z-axis, the alignment is measured by scattering 369.5 nm laser light off each of the modules' surfaces. The beam is aligned parallel to the plane of the modules and lowered onto either side of the inter-module gap. The difference in the beam height at which scatter is maximised on each of the modules is used to determine the alignment. This procedure leads to an alignment error in the z-axis of 3 μm. For comparison, 3-dimensional (3D) microfabricated ion traps constructed using wafer stacking techniques achieve an alignment accuracy between symmetric electrodes in the order of 10s of micrometers when aligned manually[34,35]. Employing precision-machined self-aligning features directly integrated within the wafers[36,37], or semi-automatic bonding processes[38], have since reduced the alignment imprecision to ≤2.5 μm. The present piezo

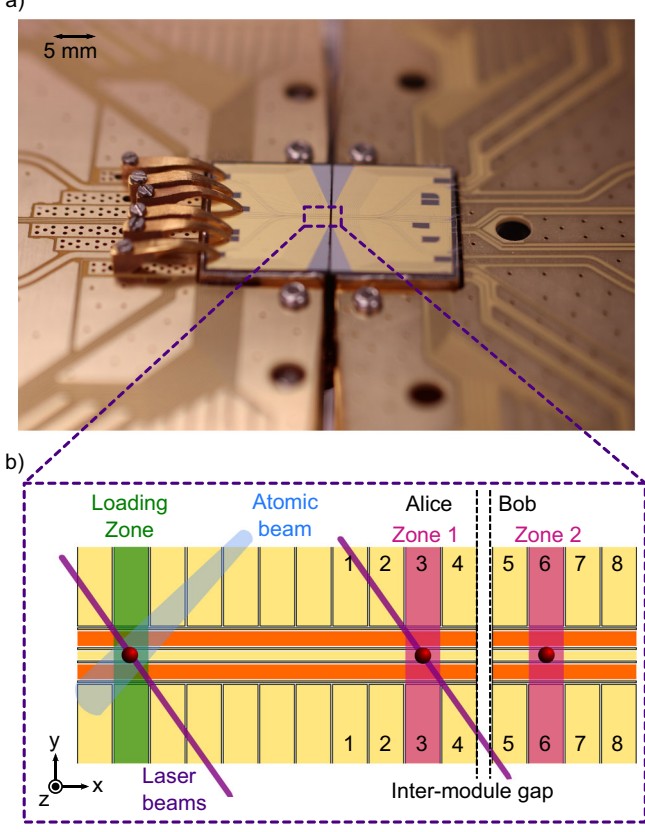

**Fig. 3 | The ion-trap microchip modules. a** Picture of the two microfabricated ion-trap modules used to demonstrate inter-module transport. Each module is 17.5 × 13.5 mm² in size. A dashed box overlay signifies the area depicted in **b**). **b** A schematic of 11 of the DC electrode pairs on the module Alice and 4 of the DC electrode pairs on the module Bob. Electrodes pairs 1–8 are used for ion transport over the 10(1) μm inter-module gap. The dimensions of the trap electrodes and inter-module gap are not drawn to scale. The DC electrodes are in yellow and the RF electrodes are in orange. Ions are loaded into the trap in the Loading Zone by the photoionisation of neutral ytterbium atoms from a hot atomic beam, whose trajectory is restricted by a metal grating. This prevents contamination of the electrodes around Zones 1 and 2. In the Loading Zone, 369.5 nm, 935.1 nm and 398.9 nm laser beams are overlapped. The 369.5 nm and 398.9 nm light is used for photoionisation, and the 369.5 nm and 935.1 nm light is used for Doppler cooling. Once an ion is loaded, the 398.9 nm light is turned off. Thereafter both the ion and the remaining laser beams are translated to Zone 1. Detection of the ion occurs in Zone 1. Zone 1 and Zone 2 form the start and end points of an inter-module link.

actuator solution therefore offers a level of alignment accuracy between separate trap modules that is comparable to the level currently reached between electrodes on stacked wafer trap designs. Following alignment of Bob with respect to Alice, we observed no measurable drift that would require periodic readjustment of the module position.

For the results presented in this work the separation between Alice and Bob in each axis is $\Delta x = 10(1)\,\mu\text{m}$, $\Delta y = 0(1)\,\mu\text{m}$ and $\Delta z = 0(3)\,\mu\text{m}$. From previous simulations[17], a misalignment in all three axes by $\leq 10\,\mu\text{m}$ should lead to a RF barrier $\leq 0.2$ meV for a trap depth of ~100 meV and an ion height of 100 μm. Overcoming an RF barrier of this magnitude has already been shown to be compatible with high-fidelity ion transport[35]. Both Alice and Bob are driven by independent RF circuits. These are calibrated to drive both modules at the same frequency, amplitude and phase (see Methods). RF voltages are applied at an amplitude $V_0 = 101.75$ V and frequency $\Omega_{RF}/2\pi = 19.32$ MHz, which from trap simulation using the Finite Element Method (FEM) yields a trap depth of 53.9 meV. FEM simulations also indicate changes in the height of the pseudopotential minimum across the inter-module gap that are limited to < 100 nm around a mean ion height of 121.97 μm.

$^{174}\text{Yb}^+$ and $^{171}\text{Yb}^+$ are used in this work. The qubit stored in the $S_{1/2}$ hyperfine manifold of $^{171}\text{Yb}^+$ is used to measure the effects of decoherence mechanisms on the matter-link, while $^{174}\text{Yb}^+$ is used to measure the infidelity associated with ion loss during transport due to its higher fluorescence rate. A detailed analysis of the isotope-dependent fluorescence and the energy level structure of $^{174}\text{Yb}^+$ and $^{171}\text{Yb}^+$ can be found in Ref. [39].

To initialise the system, isotope selective loading occurs on Alice in the Loading Zone which is shown in Fig. 3b. Once loaded, the ions are shuttled from the Loading Zone to Zone 1 over a distance of 1840 μm, which is the starting point of all subsequent experiments. The axial trap frequency is $\nu_{ax} = \omega_{ax}/2\pi = 141(1)$ kHz and the radial frequencies are $\nu_{rad} = \omega_{rad}/2\pi = 1.15(3)$ and 1.31(3) MHz. The details of the voltage control system can be found in Methods.

## Inter-module ion transport

Ion transport between modules is implemented by varying the voltages applied to the 4 electrode pairs closest to the inter-module gap on both Alice and Bob (1–8 in Fig. 3b). Successive voltage updates sent to each electrode realise a translating potential well at the ion (see Methods). Each ion transport from Zone 1 → Zone 2, or Zone 2 → Zone 1 constitutes a single matter-link between modules. Zones 1 (2) were chosen as the start or end point of the link, since the ion can be confined independently on Alice (Bob) without requiring potentials from the neighbouring module. Zones 1 and 2 are separated by a distance of 684 μm. As an initial verification step, the success of the inter-module link was confirmed by imaging the scattered ion fluorescence in Zone 1 and in Zone 2, before and after ion transport. Thereafter, the lasers and detection optics were repositioned to detect ion fluorescence in Zone 1.

The infidelity associated with ion loss during inter-module transport was measured by transporting a single $^{174}\text{Yb}^+$ ion between Zone 1 and Zone 2. After each set of $2 \times 10^5$ links the presence of the ion was verified by the detection of fluorescence using a photomultiplier tube (PMT). With a single link duration of 412.5 μs, at an equivalent link rate of 2424 s$^{-1}$, $15 \times 10^6$ consecutive links were completed successfully. The ion was subsequently lost after completing a larger, number of transport operations in the following set of transport operations, thereby placing an upper limit of $7 \times 10^{-8}$ on the infidelity associated with ion loss during transport. The ion travelled 10.26 km at an average transport speed of 1.66 m s$^{-1}$. Throughout these transport measurements, the digital-to-analogue converters (DACs) were updated at the fastest possible rate. This led to distortions in the transport waveforms, resulting from the low cut-off frequency of the DC filtering circuits. No difference in the ion lifetime for stationary and transported ions was identifiable, therefore the infidelity associated with ion loss

during inter-module transport was not measurably affected by the distortions in the DC waveforms. The limit on the probability of successful transport is therefore attributed to laser instability and ion loss from collisions with background gas molecules.

The ion-transport rate was limited by hardware constraints. Faster transport times can be achieved by using DACs with a faster update rate, modifying the DC filter circuits to have a higher cut-off frequency (see Methods) or by pre-distorting the DC waveforms to compensate for the DC filter circuits.

## Preserving qubit coherence

To show that the coherence of the qubit can be maintained throughout the matter-link, the effect of the inter-module transport on qubit states is investigated. Here the qubit is formed using two hyperfine levels of $^{171}\text{Yb}^+$ in the $S_{1/2}$ manifold: $|0\rangle \equiv |F = 0, m_f = 0\rangle$, $|1\rangle \equiv |F = 1, m_f = 0\rangle$. The two states are separated by $12{,}642{,}812{,}118 + 311 B^2$ Hz where $B$ is the magnetic field in Gauss[2]. The ambient magnetic field at the qubit is 10.177(1) G as measured on the $|F = 0, m_f = 0\rangle$ to $|F = 1, m_f = -1\rangle$ transition. The first order magnetic field insensitivity of the qubit (compared to the $|F = 1, m_f = \pm 1\rangle$ states) increases its robustness against decoherence from ambient magnetic field fluctuations with the qubit transition frequency experiencing a magnetic field dependent linear slope of 6.2 kHz G$^{-1}$ at an ambient magnetic field of 10 G. The magnetic field fluctuations are therefore expected to result in a frequency shift of up to ±6 Hz, on timescales longer than the time taken for a single fringe measurement, as indicated by near unity fringe contrast in the measurements in Fig. 4a.

A Ramsey-type experiment is used to probe the coherence of the qubit by measuring the $T_2^*$ time. This experiment is performed by first optically pumping the ion into the $|0\rangle$ state and subsequently applying two $\pi/2$ Ramsey pulses, separated by a delay time $\tau$. The pulses are applied using resonant microwave fields from an external microwave horn. The probability of the qubit being in $|1\rangle$ is then read out using a state-dependent fluorescence detection scheme[40]. The experiment is then repeated with inter-module transport operations taking place during the delay time $\tau$ (see Methods). Figure 4a shows an example of a stationary Ramsey experiment in comparison to results using 2 and 100 links within the delay time. The Ramsey fringe contrasts measured were 0.96(2), 1.00(2) and 0.97(2) for 0, 2 and 100 links respectively. The measured contrasts indicate that there is no measurable loss of qubit coherence during inter-module qubit transport for $\tau = 100$ ms.

Figure 4a shows phase offsets of 1.8690(1) rad and 3.7988(1) rad for the 2 and 100 links, respectively, relative to the stationary measurement. The experimental set-up does not include any magnetic field shielding or any active magnetic field stabilisation. These phase offsets are attributed to miscalibrations in the energy level splitting frequency of the qubit, uncompensated magnetic field drifts and transport of the qubit through spatial magnetic field inhomogeneities. Once shielding and/or active magnetic field stabalisation are installed, magnetic field drifts can be arbitrarily reduced. Any phase accumulation resulting from ion transport across quasistatic magnetic field inhomogeneities can be calibrated and compensated for using an additional phase rotation after the transport operation.

Imperfections in the electrode-voltage signal chain, such as signal distortions from the filter circuits can lead to heating. Therefore, for the Ramsey sequence an inter-module transfer rate of 1250 s$^{-1}$ was used and the number of links executed by the qubit within the delay time was restricted to 100. This ensures that the measurement statistics of the bright state remain unaffected by the reduction in fluorescence resulting from kinetic energy gain induced by ion-transport.

Figure 4a demonstrates that, within the available measurement accuracy, qubit coherence is unaffected by inter-module transport at $\tau = 100$ ms. To investigate that qubit coherence is maintained throughout transport operations between QC modules more generally, the Ramsey-type experiment shown in Fig. 4a was reproduced with longer delay times up to $\tau = 500$ ms. A Gaussian decay is then

a)

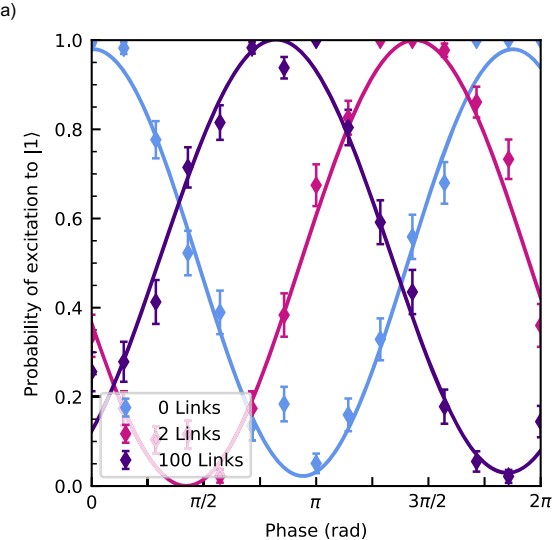

b)

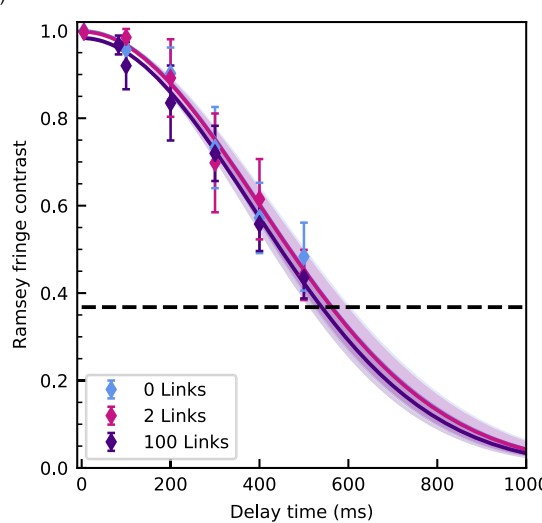

**Fig. 4 | Ramsey interferometry. a** Ramsey fringes measured after 0, 2 and 100 links. The solid lines represent a sinusoidal fit to the data. For each dataset $\tau = 100$ ms, 100 averages were taken per data point. Error bars are standard deviations. The Ramsey fringe contrasts are found to be 0.96(2), 1.00(2) and 0.97(2) for 0, 2 and 100 links respectively. **b** The Ramsey fringe contrast as a function of time for 0, 2 and 100 links. The error bars represent the standard deviation in the measured contrast for a given time delay. The Gaussian fit to each dataset is given by the solid lines and the associated shaded areas represent the 1-sigma error in the fit. The dashed line indicates the $1/e$ decoherence threshold. The coherence times are 560(40) ms, 560(40) ms and 540(30) ms for 0, 2 and 100 links respectively.

fitted to the fringe contrast to calculate a coherence time $T_2^*$, where the Gaussian decay in fringe contrast is indicative of low frequency noise dominating the dephasing process[41]. Figure 4b shows the coherence measurements and the resultant fits. For the stationary case, $T_2^* = 560(40)$ ms. With 2 links, $T_2^* = 560(40)$ ms, and with 100 links, $T_2^* = 540(30)$ ms. The main limiting factors of the coherence time are expected to be magnetic field fluctuations over the timescale of the experiment. Each of the 1-sigma errors of the measured coherence times overlap, demonstrating that we cannot detect a loss of coherence due to inter-module transport within the uncertainty of our measurement.

Figure 4 a can be used to determine an upper bound for the loss of coherence of the qubit during transport. The lower bound of the fringe contrast for 100 links is 0.95. This translates to an upper bound on the infidelity per link of $5 \times 10^{-4}$. This confirms that the matter-link can be implemented to facilitate information transfer with a high quantum-state fidelity. The actual infidelity per link is expected to be orders of magnitude lower as shown in the experimental demonstration for linear ion transport by Kaufmann et al.[16].

## Discussion

The techniques used in this work demonstrate that inter-module ion transport is a practical approach to interface QC modules. Two neighbouring surface-electrode ion-trap modules were connected using ion-transport operations, realising a fast, deterministic and high-fidelity quantum matter-link. This method of linking modules naturally extends the QCCD architecture from one to multiple modules. This demonstration will therefore facilitate the implementation of a scalable QCCD architecture capable of fault-tolerant utility-scale quantum computation. Furthermore, the inter-module link realised here is three orders of magnitude faster than the measured $T_2^*$ time. We also note that modular QCCD implementations involving different fabrication processes or ion trap designs, such as 3D microfabricated ion traps, could be connected with this method. To allow for successful multi-qubit operations following a link, it is necessary that ion transport operations with low motional excitation are realised between modules. These have already been experimentally demonstrated within linear sections of trapped ion modules where trapped ions were transported over the timescale of few secular oscillations, with sub-quanta

motional excitation[42,43]. Ion transport with low motional excitation is critical in order to maximise the fidelity of multi-qubit operations after transport events. Therefore, transport operations that facilitate inter-module and intra-module transport must be constructed to minimise motional excitation or must be combined with sympathetic cooling mechanisms at a cost of increased experiment time. Following upgrades to the filtering circuitry as well as the control hardware, future work will focus on implementing and characterising transport with sub-quanta motional excitation on the two-module system. Implementing this technique then holds the potential for an additional order of magnitude increase in the inter-module shuttling rate.

## Methods

### Effective connection rates for photonic interconnects

Scalable optical components are important for the implementation of a large-scale QC using photonic interconnects. Large switch arrays at trapped-ion wavelengths in the ultra-violet have yet to be developed, and high branching ratios in the repumper transition typically make it inefficient to use longer wavelength trapped ion transitions in the 700–900 nm range.[44] Alternatively, a large array of telecom wavelength switches combined with an ion-to-telecom wavelength conversion scheme could be used. Telecom wavelength switches with low optical losses of 2.1 dB on average have been demonstrated[45]. Furthermore, conversions from $Sr^+$ or $Yb^+$ wavelengths to the telecom bands have been shown with conversion efficiencies of ~9%[46,47]. Therefore, assuming a raw entanglement rate $R$, frequency up-conversion results in an up-converted rate of $0.09R$. Switch losses would then reduce this rate to $0.06R$. In addition, a distillation process may be required to achieve an entanglement fidelity within the fault-tolerant threshold. By following the rate calculation outlined in Ref. [48], which compares the raw entanglement rate of photonic interconnects to a distilled entanglement rate, that produces entangled states of higher fidelity, a fidelity of 99.7% could be achieved from the current raw entanglement fidelity of 94%, at the cost of reducing the effective entanglement rate. Assuming all processes except the production of entangled pairs via remote entanglement are instantaneous and assuming a mixed-species gate can be achieved with 99.9% fidelity, the effective entanglement rate would be reduced by a factor of 6. This results in a distilled entanglement rate of $0.01R$. To complete the

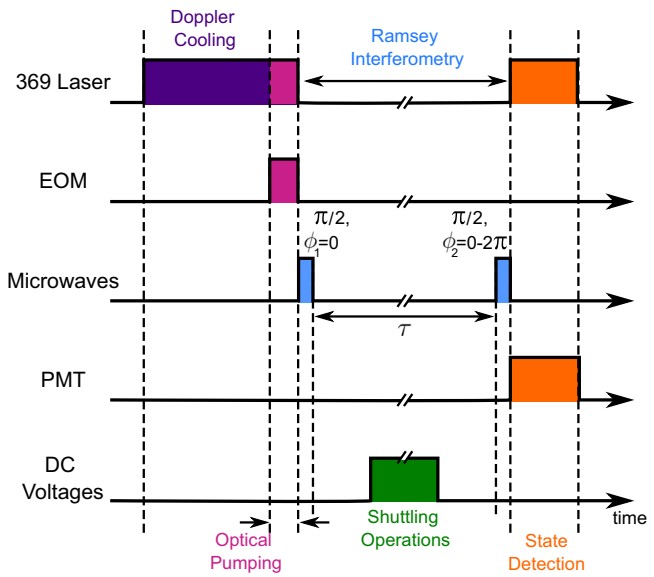

**Fig. 5 | Pulse sequence diagram for the Ramsey-type experiment.** The relative times of the different processes are represented on the same time axis (not to scale). From the top down the axes are: the on/off times for the 369.5 nm Doppler cooling laser, the electro-optic modulator (EOM) for optical pumping, the microwaves resonant with the qubit transition, the PMT for detection and the DC system for shuttling.

information transfer between two modules an additional set of gates is required as can be seen in Fig. 1a. The time taken for this process is also assumed to be instantaneous.

## Surface-electrode ion-trap modules

Within each module, electrodes were produced using standard photolithography techniques. Dicing and isotropic silicon etching steps were subsequently used to create a precision alignable edge. This combination of processes ensures that the level of fabrication accuracy and surface quality is uniform across the microchip module. The roughness of the electrode edge sidewalls is measured to be submicron and does not hinder module-module alignment capabilities. Each module features a pair of RF electrodes that are 270 µm wide, 1 µm thick and are separated by 90 µm, resulting in the RF pseudopotential capable of radially confining ions at a height of 122 µm above the module's surface. Confinement in the axial direction is provided by outer segmented static voltage electrodes with width 220 µm, thickness 1 µm and separated by a gap of 10 µm. The RF and DC electrodes are linearly cut at the inter-module interface in line with the approach developed by Lekitsch et al.[17] to connect QCCD modules via ion transport operations. To prevent electrical discharges between electrodes, the separation between the RF electrodes and surrounding DC electrodes is at least 5 µm.

## Ramsey experimental sequence

A schematic of the Ramsey-type experimental sequence can be seen in Fig. 5, and can be broken down as follows: The ion is initially Doppler cooled for up to 50 ms. Thereafter the ion is optically pumped into $|0\rangle$ over the course of 10 µs. The optical pumping is followed by an on-resonance microwave $\pi/2$ pulse with phase $\phi_1 = 0$ rad. The qubit is left to freely precess for a time delay $\tau$, before a second $\pi/2$ pulse with phase offset of $0 \le \phi_2 \le 2\pi$ rad is applied. For each phase offset, the measurement is repeated 100 times. To measure the coherence time, the experimental sequence was repeated for $\tau = \{5, 100, 200, 300, 400, 500\}$ ms for the stationary and 2 link data whereas the 100 link data spanned $\tau = \{83, 100, 200, 300, 400, 500\}$ ms. When investigating the impact of the matter-link on the $T_2^*$ time, a variable number $N = \{2, 100\}$ of qubit transport operations can be undertaken within the delay time $\tau$, such that $NT_L < \tau$, where $T_L$ is the time taken for one link (800 µs).

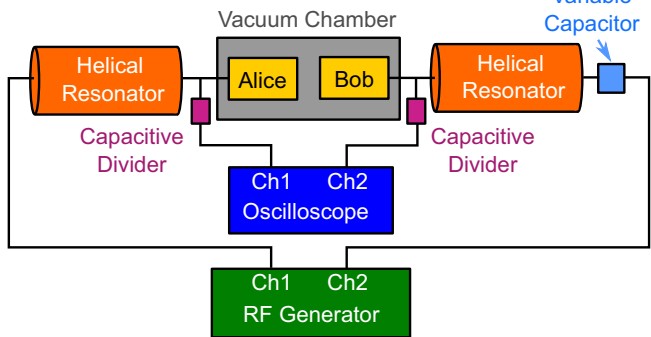

**Fig. 6 | Diagram of the RF delivery system for the Alice and Bob trap modules.** Each of the modules uses a separate RF source which is amplified and impedance matched to the ion traps using a helical resonator. A capacitive divider is used to measure the RF amplitude and phase applied to each trap on an oscilloscope. A variable capacitor is used to match the resonant frequency of the RF circuit for Bob with the RF circuit for Alice.

## Trap RF set-up

A diagram of the trap RF delivery system is shown in Fig. 6. Both modules are driven using separate direct digital synthesiser (DDS) channels from an AD9910 Urukul card. Each DDS output is amplified and a helical resonator, which has been designed following the methods presented in Ref.[49], impedance matches the RF source to the ion trap module. While the helical resonators are constructed to be identical, a variable capacitor in the RF circuit of which Bob is a part of allows for the fine tuning of the resonant frequency of the Bob RF system to match that of the Alice RF system. A 1:54 capacitive divider is used after each helical resonator to measure the signal amplitude at which the module is driven on an oscilloscope. The RF output can then be modified such that the amplitude at each module is the same within the 130 mV error of the oscilloscope. In addition, the capacitive dividers are also used to measure the relative phase of the RF signals of each of the modules and is corrected to match to within 35 µrad. The capacitive coupling between Alice and Bob is on the order of 1 fF such that parasitic cross-coupling is found to be negligible at the RF drive voltages used in this work. The absence of significant capacitive cross-coupling is further confirmed by the lack of variation in both the S11 parameter of the helical resonator, and the resonant frequencies of the RF circuits, when the modules are aligned to one another.

## DC ion transfer protocol

A Sinara Kasli field-programmable gate array (FPGA) controller equipped with three AD5372 digital-to-analog converter (DAC) Zotino cards is used to control the DC waveforms applied to the ion-trap modules, for ion transport. The DAC cards update at a rate of 139 kHz per channel. Each DAC channel has an internal third-order Butterworth filter with 75 kHz cut-off frequency. In addition, a second set of second-order RC filters with a 47 kHz cut-off frequency is used prior to the vacuum chamber. Inside the vacuum chamber each DC channel is connected to a final first-order RC filter with a 257 kHz cut-off frequency.

The ion transfer waveforms are numerically determined from a FEM electrostatic simulation. The simulation includes electrode potentials for both ion-trap modules. In order to calculate the potentials, a sequential least-squares programming (SLSQP) optimiser is used (implemented using the wrapper of the algorithm in Ref.[50] in the SciPy python library) to minimise a cost function for a given ion position. The minimisation problem is constructed from a cost function, which aims to minimise the sum of the squares of the voltages across all electrodes. In addition, two constraint functions are also aimed to be minimised. These consist of: 1. the electric field at the ion position, and 2. the axial electric field curvature (and thus the axial

secular frequency). Each constraint function is weighted with penalty factors $p_1$ and $p_2$ respectively such that the parameter that is to be minimised is normalised. This is due to several orders of magnitude in difference in the numerical values of each of these parameters. These penalty factors remain constant throughout the optimisation.

Using this method, a set of voltage values is calculated. For the simulations used in this work, potentials were calculated in 2 μm steps between Zones 1 and 2. This provides trapping potentials that are linearly incremented along the ion transport path. The evolution of the voltage on each electrode is further post-processed using a second-order Savitzky–Golay filter[51] with a moving filter window of 25 voltage values. This post-processing removes numerical noise resulting from non-optimal solutions of the minimisation problem, without distorting the waveform.

From the post-processed waveforms, the sets of voltage solutions were down-sampled to provide 12 μm incremented solutions for ion transport. This axial separation between steady state potential minima in the shuttling sequence was found to provide the best trade-off between shuttling rate and shuttling fidelity. Each set of voltages is applied at a constant time delay relative to the previous set of voltages. The combination of constant spacing and constant update rate of the voltage solutions leads to a constant velocity of the trapped ion during transport.

## Data availability
The data that support the findings of the study are available from the corresponding authors upon request.

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

## Acknowledgements

The authors thank David Bretaud for providing technical assistance with the DC filtering system and Christophe Valahu for discussions with the planning of the Ramsey experiment. Ion-trap module microfabrication was carried out at a number of facilities including the Center of Micro-NanoTechnology (CMi) at the École Polytechnique Fédérale de Lausanne (EPFL), the London Centre for Nanotechnology (LCN), and the Scottish Microelectronics Centre (SMC) at the University of Edinburgh. This work was supported by the U.K. Engineering and Physical Sciences Research Council via the EPSRC Hub in Quantum Computing and Simulation (EP/T001062/1), the U.K. Quantum Technology hub for Networked Quantum Information Technologies (No. EP/M013243/1), the European Commission's Horizon-2020 Flagship on Quantum Technologies Project No. 820314 (MicroQC), the U.S. Army Research Office under Contract No. W911NF-14-2-0106 and Contract No. W911NF-21-1-0240, the Office of Naval Research under Agreement No. N62909-19-1-2116 and the University of Sussex. F.B. acknowledges the support from the Engineering and Physical Sciences Research Council (EP/S021582/1) via the Centre for Doctoral Training in Delivering Quantum Technologies at the University College London. F.R.L.-G. acknowledges the support from the Luxembourg National Research Fund (FNR) (Project Code 11615035). S.A.K. acknowledges the support from the Engineering and Physical Sciences Research Council (EP/SO23607/1) via the Centre for Doctoral Training at the University of Bristol.

## Author contributions

M.A., F.B. and F.R.L.-G. contributed equally to this work. W.K.H. and S.W. conceived the work. F.R.L.-G. and N.I.J. designed the experiment. F.R.L.-G., N.I.J., M.A. and F.B. constructed the experimental apparatus. S.H. and M.S.-B. designed and fabricated the ion-trap modules. F.B., M.A., S.J.H. wrote the experimental control software. S.J.H., F.B., M.A., M.S.-B. produced the inter-module shuttling waveforms. M.A. and F.B. performed the experiments and analysed the data with assistance from S.A.K.. M.A., F.B. and F.R.L.-G. wrote the manuscript, which was discussed, along with its results, amongst all authors.

## Competing interests

The authors declare the following competing interests: M.A., F.B., F.R.L.-G., S.W. and W.K.H. are associated and/or hold shares with quantum computing company Universal Quantum Ltd. that will make use of some of the findings of this article in the quantum computers they develop. The remaining authors declare no other competing interests.
