## [Peer Review File · Nature Communications]

A high-fidelity quantum matter-link between ion-trap microchip modulesEditorial Note: This manuscript has been previously reviewed at another journal that is not operating a transparent peer review scheme. This document only contains reviewer comments and rebuttal letters for versions considered at *Nature Communications*.

REVIEWER COMMENTS

Reviewer #1 (Remarks to the Author):

This reviewer report is in regards to a second review of the manuscript "A high-fidelity quantum matter-link between ion-trap microchip modules" by M. Akhtar et al.

I thank the authors for their detailed and thoughtful responses to the issues raised in the previous version of this manuscript. I agree that the work has been substantially improved by these revisions.

My only lingering concern is the justification for omitting the motional excitation accrued during transport between the two trapping modules. The authors write "From a topological view, the ion transport between adjacent modules is not different to ion transport within an ion trap array." They then go on to cite previous demonstrations of low-excitation shuttling within a trap array contained in a single module. It seems to me that the investigation of potential differences between operating with multiple modules as opposed to within a single module lies at the heart of this manuscript. For example, the measurements of the internal state coherence following transport are largely motivated by the desire to show the robustness of the coherence to transport between modules. Given that significant heating during transport was observed, as shown by the limits placed on the number of links that could be achieved, measurement of the magnitude of the heating would increase the impact of this manuscript significantly.

Having said this, the manuscript in its current form is still a valuable addition to the literature and will be useful to others in the field who seek to construct larger ion-trapping systems comprised of multiple modules.

Reviewer #3 (Remarks to the Author):

I am grateful to the authors of this manuscript for respectfully responding to my comments on their previous revision. After reviewing their responses to my initial comments, I can confirm that this work is original and of interest to the scientific community concerned with engineering trapped-ion based quantum computers. Further, the manuscript is clear, demonstrates the use of proper methods and contains sufficient detail for the experiment to be reproduced. However, I regretfully am not yet convinced that the impact of this work is high enough to merit publication in *Nature Communications*.

My principal concern is that the experimental progress demonstrated here is too incremental for *Nature Communications*. I would expect a manuscript in this journal to at least describe the solution to a critical problem obstructing a larger scientific goal. While the high efficiency inter-chip transfer demonstrated may be important in the long term, I don't think the authors sufficiently justify that it is critical to scaling trapped-ion quantum computers (TIQCs). As far as I know, the utility of shuttling-based TIQCs are not currently limited purely by number of qubits but more so by gate errors and the long time required to re-cool shuttled ions - two considerations that are largely left out of this manuscript. As such, this paper would be stronger if it placed the technological development they demonstrate in a broader context and addressed the interleaving experimental concerns that would determine if this work is really going to change the course of TIQC design.

For example, the authors rely heavily on the vision of Ref. 16 (Lekitsch et al.) to argue the promise of their apparatus, which calls for implementation of tessellated ion trap modules capable of driving laser-free gates. Yet not only can the authors not give assurances about the feasibility of tessellation

beyond two chips, but the authors also do not discuss whether or not the necessary additional control lines required for the gates mentioned in Ref. 16 can be integrated into their new design. Nor do they discuss the issue of motional heating, which is already a serious limitation to shuttling-based trapped ion quantum computers and could be worse in their scheme.

Additionally, the authors of this manuscript are not able to rigorously define the potential problems associated with phase offsets potentially arising from field inhomogeneities at the chip boundary due a lack of magnetic shielding, a problem that will become more critical if their strategy were to be combined with the magnetic-field-sensitive qubits required for the laser free-gates mentioned in Ref. 16. It may be that this manuscript can stand on its own without relying fully on the scaling arguments of Ref. 16, but in that case I think it should be clearly stated that the number of qubits on a single chip is not currently a limiting factor for the utility of TIQCs and the conditions under which the number would become critical should be outlined.

While I appreciate the authors' detailed correction of my incorrect statement regarding the connectivity of the shuttling, I still think that the comparison between the two scaling strategies is too strongly stated. It has been ten years since diabatic shuttling with minimal motional excitation has been demonstrated and, as far as I know, it has still not been successfully combined with the implementation of programmable quantum gates (excepting very recent results which hint at it). But given how little successful adoption of this technique there has been, is it fair to completely disregard the amount of re-cooling time that is typically associated with shuttling in making a comparison to photonic scaling? In other words, it seems that in calculating the shuttling-based entanglement rates, the authors assumed mostly fundamental, rather than technical limitations. But in calculating the photonic entanglement rates, they only considered rates which have already been achieved, which include technical limitations of various kinds which may be alleviated in the future.

On a related note to that above, while it makes sense to focus on the contrast between photonically networked QCs and shuttling-based TIQCs, it should be pointed out that these are not the only proposals out there. For example, PRX Quantum 3, 010344 can be cited as an additional alternative.

Finally, I would strongly encourage the authors to reconsider the referees' comments regarding the term fidelity. Although I find the lengthy justification in their response to be compelling, I would worry that other readers of this manuscript - who will not have read this justification - will have the same negative response as all three referees did. Therefore, if the authors do not choose to change this wording, they will run the risk of other readers dismissing their work.

Reviewer #4 (Remarks to the Author):

The manuscript presents a first experimental implementation of shuttling of a single ion between two adjacent, but independent, trap modules. The Authors present convincing data that they can reliably transfer the ions between the two modules providing an alternative to photonic interconnects in the context of scaling trapped ion quantum computers (TI QCs). They additionally prove that during transport the coherence of the qubit state encoded in the hyperfine states of the ion is maintained, a fundamental requirement for QCs. The results are a potentially significant step forward toward the scaling of TIQCs, although not in the sense of large-scale quantum networks, but currently, or at least to my knowledge, lack other interdisciplinary applications aside from TI quantum computing.

Comments:

1) page 1 line 22, I still think that the number of nines in the transport fidelity is excessive. If the transport in the future is characterized even better and more nines are added it would become half a line long.... I think that the error rate already achieved is sufficiently high that one needs to start presenting it in terms of error rate ($\sim 7e-8$) instead of pure percentage as has been done so far.

2) page 2, line 2, the Authors state "modular QCs capable of hosting millions of trapped-ion qubits". I believe that millions is strongly misleading. Following the explanation of the authors in page 3, lines 2-10, at least 39 modules would be required to reach 10^6 qubits, twice the number for the "millionS". Assembling that many modules together is intrinsically another challenge in itself. I would suggest changing it for "large amounts of" or similar. I feel like "millions" is a too quantitative value at this stage, although this result is indeed valuable and points in the direction of a larger qubits number than the ones currently available.

3) page 2, line 9 "physical implementations 2–7." Typo on "physical". Although from the gate approach experience of the authors it is clear the reason for the citation of the work from Srinivas et al., I feel that it would be fair to also add the current highest fidelity two qubit gate implemented with lasers on trapped ions: Clark et al. Phys. Rev. Lett. 127, 130505.

4) page 4, lines 19-20 "Furthermore, for a QC architecture based on quantum matter-links, multi-species gates are not necessary." I do not think this statement is necessarily true how it is put right now, it might be necessary for state readout or sympathetic cooling in some experimental schemes. I think that what the Authors want to say is that no additional ion species is required to transfer the information itself, i.e. they refer to the transport not to the architecture. It should be written something as "Furthermore, for transport of information in a QC architecture based on quantum-matter links,..." were the correlation between the transport and the need of a secondary species is explicit.

5) page 5, line 10-12. I think a useful addition to the sentence would be the distance that separates the two transport points.

6) page 9, lines 26-27 "The ion travelled 10.26 km at an average transport speed of 1.66ms^{-1} ." Has the speed been kept constant within the single transport link instead of being adiabatically ramped up and down? i.e. if the ion went from zero to cruise speed that's an instantaneous acceleration which might as well explain the high motional excitation of the ions after the transport observed by the Authors. If the speed was actually constant I think the Authors should write it in the main text or as a minimum in the methods section.

7) page 10, line 4 "The ion-transport rate was limited by hardware constraints." Is that true? This is partially a curiosity question from my side. Has predistortion of the DC transport signal been tried? It would allow to compensate for a part of the hardware limitation, i.e. DC filtering, until hitting the physical limits of the maximum voltage on the DAC board.

8) page 12, line 1-4 I think the sentence as is right now is not extremely useful in explaining it as a limitation in increasing the number of transports. As it is phrased right now it still feels like an arbitrary choice done for optimization and not a causal relation that fundamentally limits the experiment. It would be more useful to state that a larger number of links was not implemented due to the heat induced by the transport which in turn affected the fluorescence of the ion. Making this clear would help other readers to also understand why heat gained per transport has not been measured since they are related. I would suggest something like "... Ramsey sequence was limited to 100, due to the large amount of heat gained in the transport affecting the fluorescence." or similar.

9) page 14 section "Surface-electrode ion-trap modules". No detailed information is provided on the specific structure of the RF rails at the interconnection point between modules aside from "precision alignable edge", even the main text is vague. What is the shape of the electrodes at the endpoint of the rails? From my understanding it should be a simple linear cut as in Lekitsch et al or fig 2)c. Additionally what is the distance from the end of the electrodes to the end of the substrate/beginning of gap? This distance might be critical for people interested in reproducing the results, is it effectively zero? I also think that this section should present a picture (SEM preferably or if unavailable a schematic in scale of the actual trap section) of the end rails to allow readers to actually visualize how the electrodes ends at the interconnection. Fig 2c) is deemed an illustration and not representative of the actual experimental conditions reported.

10) page 15 line 9, I suggest adding again here the link time (412.5 us) to improve understanding.

11) page 16, line 5-7 "The axial trap frequency is $\nu_{ax} = \omega_{ax}/2\pi = 141(1)$ kHz and the radial frequencies are $\nu_{rad} = \omega_{rad}/2\pi = 1.15(3)$ and $1.31(3)$ MHz." I feel that this information is important enough to be added to the main text. I was effectively looking for it since motional mode frequencies are strictly connected to ion transport.

12) page 16, line 11-12 "The cost function is constructed to fulfil the following criteria:" the information on the cost function is vague, what is the relative weight for the three subcost function? Have they been simply summed? Sum of the exponentials? I think more mathematical details are required to be able to reproduce it given that the transport is at the heart of the subject of the manuscript.

In regard to comments of the previous Referee #2:

Mostly the comments have been addressed in a satisfying manner. There are only a few points left in my opinion:

B) I think the newly introduced discussion from page 6 lines 13-21 is well written, but still lacks a physical explanation for how the drifts might arise for the non-experts, i.e. why are they actually expected to happen.

D) I think this comment has been discussed by the authors, but my own personal opinion still differs, see my comment number 1).

6) The new detailed data on the trap structure is indeed very useful, I think that a useful information for reproducibility that is still missing would be the thickness of the electrodes.

9) I think this comment has not yet been properly addressed, see my own comment number 8).

We are grateful to the referees for the time they have dedicated in reviewing our revisions to the manuscript. We have addressed all comments in the detailed responses below and made appropriate amendments to the manuscript. With the accompanied changes, we hope our revised manuscript can be published now.

Referee #1 (Remarks to the Author):

The referee states: “My only lingering concern is the justification for omitting the motional excitation accrued during transport between the two trapping modules. The authors write "From a topological view, the ion transport between adjacent modules is not different to ion transport within an ion trap array." They then go on to cite previous demonstrations of low-excitation shuttling within a trap array contained in a single module. It seems to me that the investigation of potential differences between operating with multiple modules as opposed to within a single module lies at the heart of this manuscript. For example, the measurements of the internal state coherence following transport are largely motivated by the desire to show the robustness of the coherence to transport between modules. Given that significant heating during transport was observed, as shown by the limits placed on the number of links that could be achieved, measurement of the magnitude of the heating would increase the impact of this manuscript significantly.”

We agree with the referee that motional heating is a significant factor when considering the implementation of shuttling-based quantum algorithms on ion trap devices. However, accurate and meaningful measurements of the acquired kinetic energy following inter-module ion transport could not be included as the set-up cannot currently be used to implement sideband thermometry techniques. This is due to the current lack of an on-chip gradient to allow for long-wavelength spin-motion coupling because of a broken wire connection that connects the chip which needs to be reconnected inside the vacuum system. This connection will be repaired in the next set of changes made to the experiment. Following these changes, the aim will be to characterise and minimise the motional heating accumulated during transport. However, due to the extensive nature of the changes required, adding this information to the current manuscript is not possible.

Nevertheless, we would like to point out that the considerations for intra-module and inter-module shuttling are fundamentally very similar.

Firstly, while a key difference between a two-module set-up and a single device resides in that a two-module set-up has disjoint RF electrodes, the underlying physical mechanism of this situation

is the same as for a single device featuring an X-junction. Indeed, the RF electrodes' discontinuity at the inter-module gap between modules gives rise to a small RF pseudopotential barrier. Transporting ions across an RF pseudopotential barrier with minimal motional excitation has already been successfully demonstrated (see Blakestad *et al.*, Physical Review Letters, 102, 153002 (2009)). Furthermore, as with an X-junction, a trapped ion located on a non-zero gradient of the RF barrier induced by the inter-module gap will experience some heating due to the coupling to the noise on the trapping RF pseudopotential (Wineland *et al.*, J. Res. Natl. Inst. Stand. Technol., 103, 259 (1998)). In practice this can be easily mitigated by reducing the duration spent at this location, i.e. by increasing the transport rate.

Secondly, for both intra- and inter-module transport, kinetic energy gains are highly dependent on the transport operation that is used. There, variations in the transport voltages will impart a different number of phonons onto the trapped ion's final state of motion. For quantum information processing, transport operations should not only be realised with low motional excitation but also be fast, i.e. of the same order as the quantum logic operations. Here, the theoretical work of Simsek and Mintert, *Quantum*, 5, 409 (2021) showed that constructing appropriate shuttling voltage waveforms for the implementation of fast ground-state to ground-state transport methods is possible even the presence of an RF barrier induced by an X-junction such that a similar capability is expected for an RF barrier induced by the inter-module gap.

As we have detailed in our previous reply to the referee, while the demonstration of diabatic shuttling is not a novelty in itself, we would like to demonstrate such transport operation between the two modules in the future. However, this also remains inaccessible to us at this point in time as the set-up would have to undergo a significant upgrade while again also calling for sideband thermometry capabilities.

Nevertheless, we would like to point out that even in the event where an ion were to be subject to a transport operation through the inter-module gap resulting in a non-zero \underline{n} , this excessive kinetic energy may then be dissipated via sympathetic cooling. Such a method has already been demonstrated on a Yb-Ba ion chain (Inlek *et al.*, Physical Review Letters 118, 250502 (2017)) with ions cooled close to the motional ground-state, thereby permitting the subsequent implementation of two-qubit quantum logic gates. Alternatively, one can make use of alternative gate mechanisms which can realise very high gate fidelities without the need of ground-state cooling and as such work with high motional state excitation after ion transport, see for example, Valahu *et al.* New. J. Phys 23, 113012 (2021).

To make the above considerations clearer to the reader and emphasise the effects of kinetic energy gains in a shuttling-based QC architecture, we have added a discussion to the revised manuscript on page 13 - line 29 – page 14 lines 1-4: “Ion transport with low motional excitation is critical in order to maximise the fidelity of multi-qubit operations after transport events. Therefore, transport operations that facilitate inter-module and intra-module transport must be constructed to minimise motional excitation or must be combined with sympathetic cooling mechanisms at a cost of increased experiment time.”

Kinetic energy acquired during ion transport can always be mitigated using the techniques explained above and as such is a property of both intra- and inter-module ion shuttling and is not specific to the demonstration reported here. Therefore, we believe that the demonstrations that an ion can be transported between modules with a low infidelity associated to ion loss, and without measurable loss of coherence, to be highly important results with significant impact.

Referee #3 (Remarks to the Author):

The referee states: “My principal concern is that the experimental progress demonstrated here is too incremental for Nature Communications. I would expect a manuscript in this journal to at least describe the solution to a critical problem obstructing a larger scientific goal. While the high efficiency inter-chip transfer demonstrated may be important in the long term, I don't think the authors sufficiently justify that it is critical to scaling trapped-ion quantum computers (TIQCs). As far as I know, the utility of shuttling-based TIQCs are not currently limited purely by number of qubits but more so by gate errors and the long time required to re-cool shuttled ions - two considerations that are largely left out of this manuscript. As such, this paper would be stronger if it placed the technological development they demonstrate in a broader context and addressed the interleaving experimental concerns that would determine if this work is really going to change the course of TIQC design.”

Our manuscript addresses the problem of developing utility-scale quantum computers. Current work in quantum computing focusses on devices that incorporate 50-100 qubits. Experiments are often carried out in the NISQ regime where the inherent errors of quantum operations are not corrected. Very few applications are known for the NISQ regime. Most well-known applications for quantum computing require the operating regime of fault-tolerant quantum computing where inherent errors are corrected making use of large numbers of extra qubits. This type of operating mode is required by most disruptive quantum computing applications. The examples of simulating the FeMoco molecule required for fertilizer production and breaking bitcoin encryption are discussed in some detail in AVS Quantum Sci. 4, 013801 (2022). From that discussion it becomes obvious that a useful quantum computer would typically require 100,000 or even millions of qubits. It is not possible to fit so many trapped ion qubits on a single wafer, so a multi-modular solution is required. In order for such a solution to be suitable for fault-tolerant quantum computing one needs to have a sufficient connection speed and transmit quantum information with sufficient fidelity. There are two distinct contributions to the infidelity. One is the decay of the quantum state. In Fig. 5 we experimentally verify that there is no measurable decay of the coherence of the qubit and as such this infidelity remains below the fault-tolerant threshold. Indeed Kaufmann *et al.*, Physical Review Letters 13 120, 010501 (2018) have verified in separate experiments that the quantum state of the ion is not affected by transport. The second source of infidelity consists of the possibility of ion loss during transport between modules. The second quantity has never been measured before and corresponds to the main novelty of this manuscript. Again, this infidelity is below the fault-tolerant threshold. As such, this manuscript constitutes the first demonstration of connecting two trapped ion quantum computing microchips with both fidelity and speed sufficient

for the execution of fault-tolerant quantum computing. Therefore, we believe this demonstration is not incremental and instead constitutes a significant milestone towards building utility-scale quantum computers with trapped ions.

Ion trap quantum computing has already demonstrated gate errors below the fault-tolerant threshold so trapped ion quantum computing is not limited by gate errors anymore. A demonstration of a special type of ion transport has already been carried out ten years ago, diabatic shuttling, where the ion is shuttled fast, yet the potential is switched in such a way that kinetic energy picked up during transport is mitigated (see Physical review letters 109, 080502 (2012) or Physical review letters 109, 080501 (2012)), therefore not requiring any recooling. Currently, filters installed in our experiment prevent switching voltages fast enough to realise diabatic shuttling operation. We are currently switching filters in the experimental set-up so that we can carry out an experimental demonstration in the system in the next few months and this demonstration will form part of a separate paper. We note that there is only limited novelty in this demonstration as diabatic transport has already been demonstrated by two separate research groups ten years ago. Finally, we should also mention that by making use of microwave quantum logic, it is possible to devise gate mechanisms that can generate extremely high fidelities without requiring the ion to be in the ground state. For example, New. J. Phys 23, 113012 (2021) explains a gate mechanism where Doppler cooled ions are perfectly sufficient to achieve gate fidelities past the fault-tolerant threshold.

The referee states: “For example, the authors rely heavily on the vision of Ref. 16 (Lekitsch *et al.*) to argue the promise of their apparatus, which calls for implementation of tessellated ion trap modules capable of driving laser-free gates. Yet not only can the authors not give assurances about the feasibility of tessellation beyond two chips, but the authors also do not discuss whether or not the necessary additional control lines required for the gates mentioned in Ref. 16 can be integrated into their new design. Nor do they discuss the issue of motional heating, which is already a serious limitation to shuttling-based trapped ion quantum computers and could be worse in their scheme.”

We only require tessellation of chips with an accuracy of 10 μm . Microchip processing sometimes includes alignment with precision orders of magnitude better. There are plenty of engineering procedures available to assure alignment with sufficient accuracy, for example, interferometric techniques or image analysis. As such, we believe carrying out a proof-of-principle demonstration with two modules is certainly sufficient as a demonstration of the underlying technique. For the

quantum computing blueprint discussed in Ref. 16 (now Ref. 17), the control lines are attached at the backside of the quantum computing modules via TSVs. As such these connections do not interfere with ion transport between microchip modules. A detailed explanation of how control lines are connected on the backside is contained in Ref. 16 (now Ref. 17). While we have not included backside connections with the chips being used here, the techniques to include backside connections are well understood and can be realized using standard microfabrication processes.

We agree with the referee that motional heating associated with ion transport should be discussed in our manuscript. A demonstration of a special type of ion transport has already been carried out ten years ago, diabatic shuttling, where the ion is shuttled fast, yet the potential is switched in such a way that kinetic energy picked up during transport is mitigated (see Physical review letters 109, 080502 (2012) or Physical review letters 109, 080501 (2012)), therefore mitigating kinetic energy picked up during ion transport. Additionally, even in the event where an ion were to be subject to a transport operation through the inter-module gap resulting in a non-zero \underline{n} , this excessive kinetic energy may then be dissipated via sympathetic cooling. Such a method has already been demonstrated on a Yb-Ba ion chain (Inlek *et al.*, Physical Review Letters 118, 250502 (2017)) with ions cooled close to the motional ground-state, thereby permitting the subsequent implementation of two-qubit quantum logic gates. Alternatively, one can make use of alternative gate mechanisms which can realize very high gate fidelities without the need of ground state cooling and as such work with high motional state excitation after ion transport, for example, Valahu *et al.* New. J. Phys 23, 113012 (2021).

To address the referees' comment, the following lines have been added into the manuscript on page 13 - line 29 – page 14 lines 1-4: “Ion transport with low motional excitation is critical in order to maximise the fidelity of multi-qubit operations after transport events. Therefore, transport operations that facilitate inter-module and intra-module transport must be constructed to minimise motional excitation or must be combined with sympathetic cooling mechanisms at a cost of increased experiment time.”

The referee states: “Additionally, the authors of this manuscript are not able to rigorously define the potential problems associated with phase offsets potentially arising from field inhomogeneities at the chip boundary due a lack of magnetic shielding, a problem that will become more critical if their strategy were to be combined with the magnetic-field-sensitive qubits required for the laser free-gates mentioned in Ref. 16.”

We thank the referee for their comment and we agree that magnetic field sensitive qubits, as in Ref 16. (now Ref. 17), will be more prone to phase accumulation if these are transported across magnetic field inhomogeneities. For gate interactions we will use the dressed state qubit, while for ion transport we use the clock qubit. Both are first-order insensitive to magnetic field fluctuations. Nevertheless, there will be phase pickup during ion transport. The accumulation of phase is expected to result purely from the trapped-ion qubit interaction with the static magnetic field landscape that it traverses throughout the transport operation. We would like to point out that phase accumulation is not a problem exclusive to inter-module transfer, or the chip boundary itself, but shuttling-induced phase offsets are accumulated from any transfer across magnetic field inhomogeneities along the shuttling path. These constant offsets only depend on the shuttling paths, and can be accounted for in the execution of quantum algorithms simply by keeping track of the accumulated phase offsets.

The referee states: “It may be that this manuscript can stand on its own without relying fully on the scaling arguments of Ref. 16, but in that case I think it should be clearly stated that the number of qubits on a single chip is not currently a limiting factor for the utility of TIQCs and the conditions under which the number would become critical should be outlined.”

The referee is right in pointing out that there are currently several pressing challenges aside from qubit numbers that currently hamper the demonstration of a fully operational stand-alone quantum charged-couple device. However, as highlighted in our reply to the referee’s first comment, and, as we look into the future, nearly all known practical applications for quantum computers, for example, the simulation of the FeMoco molecule or breaking Bitcoin encryption would require a number of qubits in the 100,000s or millions (see Webber *et al.* AVS Quantum Sci. 4, 013801 (2022)) thus far larger than the 10,000s qubits that can be fitted within a single QCCD module. Indeed, most applications for quantum computers would require fault-tolerant quantum computing. While there is intense research to find applications for the NISQ regime of quantum computing, there is only very limited progress so far. Scaling trapped ion quantum computers based on the QCCD approach would therefore call for a modular approach. A natural way to scale up the QCCD approach is to connect multiple modules using ion transport operation as proposed in the work of Lektisch *et al.* [Ref. 16 (now Ref. 17)]. In this current manuscript, through the first successful demonstration of quantum information transfer between modules using ion transport operations our research has demonstrated one of the key features that underpins this scalable QC vision. More broadly speaking, we also feel that such a demonstration is significantly beneficial to the entirety of the quantum technology and TIQC research field much

beyond the particular vision of a microwave based trapped ion quantum computer depicted in Lekitsch et al., as our scaling approach can be applied to any quantum computing concept which requires more qubits than can fit on a single ion trap chip. As such, this demonstration is a significant milestone towards the practical realisation of a fault-tolerant quantum computer.

To present the utility of single QCCD devices and the above condition clearly to the reader, the manuscript was amended on page 3, lines 2-7 from: “Recently, QCCD architectures have been used to demonstrate key steps towards the fault-tolerant operation of a trapped ion quantum computer. To unlock many of the anticipated applications of a QC within the necessary level of error correction, far larger qubit numbers will be required than are available on current devices”, to now read: “Recently, QCCD architectures have been used to demonstrate key steps towards the fault-tolerant operation of a trapped ion quantum computer and stand-alone near-term devices offer an attractive platform for the execution of restricted noisy intermediate-scale quantum algorithms [Cerezi *et al.*, Nat. Rev. Phys., 9, 625-644 (2021) ; Bharti *et al.*, Rev. Mod. Phys., 94, 1, 015004 (2022)] and simulations [Altman *et al.*, PRX QUANTUM, 2, 1, 017003 (2021)]. To unlock many of the anticipated applications of a QC within the necessary level of error correction, far larger qubit numbers will be required than are available on current devices”

The referee states: “While I appreciate the authors' detailed correction of my incorrect statement regarding the connectivity of the shuttling, I still think that the comparison between the two scaling strategies is too strongly stated. It has been ten years since diabatic shuttling with minimal motional excitation has been demonstrated and, as far as I know, it has still not been successfully combined with the implementation of programmable quantum gates (excepting very recent results which hint at it). But given how little successful adoption of this technique there has been, is it fair to completely disregard the amount of re-cooling time that is typically associated with shuttling in making a comparison to photonic scaling? In other words, it seems that in calculating the shuttling-based entanglement rates, the authors assumed mostly fundamental, rather than technical limitations. But in calculating the photonic entanglement rates, they only considered rates which have already been achieved, which include technical limitations of various kinds which may be alleviated in the future.”

We appreciate the referee's comment on the comparison between shuttling based and photonic mechanisms to transfer quantum information between individual ion traps. We agree with the referee that each of the mechanisms will be subject to its own set of technical challenges. One of the technical challenges for a scalable device based on ion transport operations will be operating

at, or near, the motional ground state, or efficiently using sympathetic cooling techniques to maintain a low motional state of the qubit. In fact, in a multi-module system employing either ion transport or photonic mechanisms to distribute quantum information, a larger number of operations will be necessary to supplement these transfer mechanisms. These include supplementary operations such as further gates and/or shuttling operations. Due to the large number of possible individual operations that would be necessary to employ these mechanisms, the comparison we have used focuses on the process of the link only.

Here we have used a calculation previously published in the work of Nigmatullin *et al.* New J. Phys. 18 103028 (2016) to outline the current state-of-the-art and to interpret the effective connection rates between individual modules in the context of large-scale error-corrected quantum computing. We agree with the referee that minimising motional excitation is a concern in all trapped ion quantum computing implementations and should therefore be discussed. To make the technical limitations clearer in the revised manuscript we have therefore added the following on page 13 - line 29 – page 14 lines 1-4: “Ion transport with low motional excitation is critical in order to maximise the fidelity of multi-qubit operations after transport events. Therefore, transport operations that facilitate inter-module and intra-module transport must be constructed to minimise motional excitation or must be combined with sympathetic cooling mechanisms at a cost of increased experiment time.”

The referee states: “On a related note to that above, while it makes sense to focus on the contrast between photonically networked QCs and shuttling-based TIQCs, it should be pointed out that these are not the only proposals out there. For example, PRX Quantum 3, 010344 can be cited as an additional alternative.”

We are thankful to the referee for pointing this out and agree that a reference to the work by Ramette *et al.*, PRX Quantum 3, 010344 (2022) should also be included in the manuscript. We have now revised the introduction to include on page 5, lines 3-7 the statements: “By contrast, an alternative implementation relying on the entanglement of trapped ion chains through heralded single-photon transfer in shared overlapped cavities offers fast (μs) interaction rates. Nevertheless, the approach is likely to remain constrained to tens of thousands of qubits after which slower photonic links may be relied upon for further expansion.”

The referee states: “Finally, I would strongly encourage the authors to reconsider the referees' comments regarding the term fidelity. Although I find the lengthy justification in their response to be compelling, I would worry that other readers of this manuscript - who will not have read this justification - will have the same negative response as all three referees did. Therefore, if the authors do not choose to change this wording, they will run the risk of other readers dismissing their work.”

We appreciate the referee's comment on the use of the term fidelity for the transport operations. As we intend to be absolutely clear with our communication, we placed the following statement (originally located on page 9 in the transferred manuscript) regarding relevant definitions and their motivation directly to the beginning of the revised manuscript on page 2, lines 23-29 and have therefore reworded it as an infidelity associated with ion loss during transport which makes this now explicitly clear. The statement, which has been slightly amended in line with the surrounding context, reads: “For quantum information processing using a shuttling-based trapped ion architecture, there are two sources of infidelity to consider. The first source of infidelity is the decay of the quantum state during the ion transport operation. Work by Kaufmann *et al.* [Physical Review Letters 120, 010501 (2018)] has already demonstrated that ion transport operations within a single device can give rise to a state fidelity of 99.9994% per transport operation. The second possible source of infidelity is the infidelity associated with ion loss during transport.”

To avoid repetition, the line stating: “Experimental work has shown that the transport of ion qubits within a single device can be realised with high-fidelity and with indiscernible contribution to coherence loss.” directly above this paragraph has been removed.

For consistency the abstract has also been modified such that on page 1, lines 20-22, it reads in the revised manuscript: “Ion transport between adjacent modules is realised at a rate of 2424 s^{-1} and with an infidelity associated with ion loss during transport below 7×10^{-8} .” replacing: “Ion transport between adjacent modules is realised at a rate of 2424 s^{-1} and with ion-transport success fidelity in excess of 99.999993%.”.

We feel that this explanation is clear, addresses the referees' concern as well as being in line with the key precedence paper on this topic, namely, Kaufmann et al. (Physical Review Letters 120, 010501 (2018)) where the term “transport fidelity” corresponds to the probability of transporting the ion as intended.

Referee #4 (Remarks to the Author):

The referee states: “Page 1 line 22, I still think that the number of nines in the transport fidelity is excessive. If the transport in the future is characterized even better and more nines are added it would become half a line long.... I think that the error rate already achieved is sufficiently high that one needs to start presenting it in terms of error rate ($\sim 7e-8$) instead of pure percentage as has been done so far.”

We thank the referee for their comment. The authors agree that it would be more concisely communicated (and perhaps be more readily comparable for future demonstration) as an infidelity. Therefore, the ion transport success fidelity has been restated as an infidelity associated with ion loss during transport, such that the revised manuscript on page 1, lines 20-22, the abstract reads: “Ion transport between adjacent modules is realised at a rate of $2424s^{-1}$ and with an infidelity associated with ion loss during transport below 7×10^{-8} .” replacing: “Ion transport between adjacent modules is realised at a rate of $2424s^{-1}$ and with ion-transport success fidelity in excess of 99.999993%.”.

The referee states: “Page 2, line 2, the Authors state "modular QCs capable of hosting millions of trapped-ion qubits". I believe that millions is strongly misleading. Following the explanation of the authors in page 3, lines 2-10, at least 39 modules would be required to reach 10^6 qubits, twice the number for the "millionS". Assembling that many modules together is intrinsically another challenge in itself. I would suggest changing it for "large amounts of" or similar. I feel like "millions" is a too quantitative value at this stage, although this result is indeed valuable and points in the direction of a larger qubits number than the ones currently available.”

The authors agree with the referee that the phrase in question is too strongly focussed on a particular number. Instead, we wish to highlight the milestone in the abstract as a key milestone towards utility-scale quantum computers. We have therefore amended the phrase from: “...QCs capable of hosting millions of trapped-ion qubits” to “... QCs capable of fault-tolerant utility-scale quantum computation”.

The referee is correct that the alignment of 10s of modules requires a solid engineering approach for the alignment of so many modules. Our current manuscript is a proof-of-principle demonstration and as such we cannot include such detailed engineering discussions here.

The referee states: “Page 2, line 9 "physical implementations 2–7." Typo on "physical". Although from the gate approach experience of the authors it is clear the reason for the citation of the work from Srinivas et al., I feel that it would be fair to also add the current highest fidelity two qubit gate implemented with lasers on trapped ions: Clark et al. Phys. Rev. Lett. 127, 130505.”

We thank the referee for pointing this out. The typo (page 2, line 9 in the revised manuscript) has been corrected and the additional reference (Ref. 8, also used on page 2, line 9) has also been added.

The referee states: “Page 4, lines 19-20 "Furthermore, for a QC architecture based on quantum matter-links, multi-species gates are not necessary." I do not think this statement is necessarily true how it is put right now, it might be necessary for state readout or sympathetic cooling in some experimental schemes. I think that what the Authors want to say is that no additional ion species is required to transfer the information itself, i.e. they refer to the transport not to the architecture. It should be written something as "Furthermore, for transport of information in a QC architecture based on quantum-matter links,..." were the correlation between the transport and the need of a secondary species is explicit.”

We thank the referee for indicating this and agree with their analysis. We have followed the referee’s suggestion and amended the phrase, on page 4, lines 18-20 to read: “Furthermore, for information transfer in a QC architecture based on quantum matter-links, multi-species gates are not necessary.”

The referee states: “page 5, line 10-12. I think a useful addition to the sentence would be the distance that separates the two transport points.”

The authors agree that the addition of the shuttling distance would be informative to the reader. Following this suggestion, we have amended the sentence, now on page 6, lines 9-12, to the following: “In this article, we demonstrate the transfer of trapped ions between two quantum computing modules at a transfer rate of 2424s^{-1} over a total distance of $684\ \mu\text{m}$, and with an infidelity associated with ion loss during transport of less than 7×10^{-8} .”

The referee states: “Page 9, lines 26-27 "The ion travelled 10.26 km at an average transport speed of 1.66ms⁻¹." Has the speed been kept constant within the single transport link instead of being adiabatically ramped up and down? i.e. if the ion went from zero to cruise speed that's an instantaneous acceleration which might as well explain the high motional excitation of the ions after the transport observed by the Authors. If the speed was actually constant I think the Authors should write it in the main text or as a minimum in the methods section.”

The transport voltages are applied using a quasistatic approximation of the trapped ion at each regularly spaced, discrete position in the transport waveform. Each set of voltages is also applied with constant time delay between each step. However, no considerations are being made regarding the velocity profile of the axial potential well that confines the ion during transport such that it is not adiabatically ramped up and down. For the calculation regarding the average speed, the total distance traveled for each periodic transport waveform was divided by the time taken to play back this waveform on the DAC hardware. We thank the referee for noticing this omission as it is an important part of the ion transport method. To reflect this, the Methods section on page 18, lines 24-27, has been amended to the following: “Each set of voltages is applied at a constant time delay relative to the previous set of voltages. The combination of constant spacing and constant update rate of the voltage solutions leads to a constant velocity of the trapped ion during transport.”

The referee states: “Page 10, line 4 "The ion-transport rate was limited by hardware constraints." Is that true? This is partially a curiosity question from my side. Has predistortion of the DC transport signal been tried? It would allow to compensate for a part of the hardware limitation, i.e. DC filtering, until hitting the physical limits of the maximum voltage on the DAC board.”

We agree with the referee and thank them for suggesting that pre-distortion of the DC waveforms as a method to compensate for limitations of the DC filtering, up to the maximum DC voltage. While we have not tried this technique yet, we are currently working to exchange filter circuits. We have amended the text to clarify the term ‘hardware constraints’. On page 10, line 4, the manuscript used to read: “The ion-transport rate was limited by hardware constraints. Faster transport times can be achieved by changing to DACs with a faster update rate and by modifying the DC filter circuits to have a higher cut-off frequency (see Methods).”. In the revised manuscript on page 10, lines 21-25, this now reads: “The ion-transport rate was limited by hardware constraints. Faster transport times can be achieved by using DACs with a faster update rate,

modifying the DC filter circuits to have a higher cut-off frequency (see Methods) or by pre-distorting the DC waveforms to compensate for the DC filter circuits.”

The referee states: “Page 12, line 1-4 I think the sentence as is right now is not extremely useful in explaining it as a limitation in increasing the number of transports. As it is phrased right now it still feels like an arbitrary choice done for optimization and not a causal relation that fundamentally limits the experiment. It would be more useful to state that a larger number of links was not implemented due to the heat induced by the transport which in turn affected the fluorescence of the ion. Making this clear would help other readers to also understand why heat gained per transport has not been measured since they are related. I would suggest something like "... Ramsey sequence was limited to 100, due to the large amount of heat gained in the transport affecting the fluorescence." or similar.”

We thank the referee for their comment regarding our decision to limit the maximum number of inter-module transport operations to 100 which we believe should indeed be made clearer in the manuscript. Kinetic energy gain of the trapped ion results in decreased initial fluorescence when detecting the trapped ion in the bright state, which can lead to deviations from the measured SPAM characteristics of a stationary ion. If this mechanism is not accounted for, the altered SPAM characteristics of the bright ($|1\rangle$) state could appear as an asymmetric decay in the Ramsey fringe contrast, i.e. where fringe contrast amplitudes for probabilities >0.5 are smaller than fringe contrast amplitudes for probabilities <0.5 as the stationary ion threshold is no longer a good threshold parameter to postprocess experimentally detected counts in probabilities. There are several ways to account for this. One of these is the use of “calibration runs” i.e. state preparation calibrations which interleave ion transport operations such as in Kaufmann *et al.*, Physical Review Letters 13 120, 010501 (2018). However, in this work we followed a different approach where, instead, we decided to constrain the total number of transport operations to the largest value which does not cause a measurable change in the optimal threshold value that defines the threshold between detecting a dark ($|0\rangle$) and a bright ($|1\rangle$) state.

Therefore, to communicate this better to the reader, the statement in the which used to read: “Imperfections in the electrode-voltage signal chain, such as signal distortions from the filter circuits can lead to heating. Therefore, in order to limit the kinetic energy gain of the ion, which is required for optimal qubit state detection, an inter-module transfer rate of 1250 s^{-1} was used and the upper limit of the number of links undertaken by the qubit within the delay time of the Ramsey sequence was set to 100.”, has been changed in the revised manuscript on page 12,

line 19-27, to read: “Imperfections in the electrode-voltage signal chain, such as signal distortions from the filter circuits can lead to heating. Therefore, for the Ramsey sequence an inter-module transfer rate of 1250s^{-1} was used and the number of links executed by the qubit within the delay time was restricted to 100. This ensures that the measurement statistics of the bright state remain unaffected by the reduction in fluorescence resulting from kinetic energy gain induced by ion-transport.

The referee states: “Page 14 section "Surface-electrode ion-trap modules". No detailed information is provided on the specific structure of the RF rails at the interconnection point between modules aside from "precision alignable edge", even the main text is vague. What is the shape of the electrodes at the endpoint of the rails? From my understanding it should be a simple linear cut as in Lekitsch et al or fig 2)c. Additionally what is the distance from the end of the electrodes to the end of the substrate/beginning of gap? This distance might be critical for people interested in reproducing the results, is it effectively zero? I also think that this section should present a picture (SEM preferably or if unavailable a schematic in scale of the actual trap section) of the end rails to allow readers to actually visualize how the electrodes ends at the interconnection. Fig 2c) is deemed an illustration and not representative of the actual experimental conditions reported.”

We are grateful to the referee for pointing this out and we agree that from both a scientific and reproducibility standpoint that more extensive details should be added regarding the topology of the ion trap modules at their connecting edges.

With respect to the shape of the electrodes at the end of the endpoint of the RF rails, the referee is correct that these are designed and fabricated as a simple linear cut as presented by Lektisch *et al.*, Science Advances 3, 1-12 (2017). To clarify this, we have now added a sentence in the Methods section on page 15, lines 12-14, which reads: “The RF and DC electrodes are linearly cut at the inter-module interface in line with the approach developed by Lekitsch *et al.* to connect QCCD modules via ion transport operations”.

Concerning the comment on the end of the substrate and the end of the electrode structures/beginning of the inter-module gap, each module was fabricated such that the silicon substrate recedes by $\sim 75\ \mu\text{m}$ in order to ease the alignment between the two modules. We agree with the referee that this is a useful design detail which merits to be included. We have therefore revised the manuscript on page 6, lines 17-20, to state: “Both ion-trap microchip modules have

been fabricated on a silicon substrate and feature electrode structures that extend to the edge of the inter-module gap (see Methods) while the substrate recedes $\sim 75 \mu\text{m}$ away from the gap to facilitate module alignment.”

The authors recognise the usefulness of informing the scientific community of the fabrication methods that were developed and design choices that were made to successfully realise such ion trap modules. While going in such details here would be out of the scope of this manuscript, the authors can confirm that a paper is currently in preparation and will report on the module fabrication in great detail, including detailed SEM images along with a comprehensive visualisation of the ion trap module.

The referee states: “Page 15 line 9, I suggest adding again here the link time (412.5 us) to improve understanding.”

We thank the referee for this suggestion and agree that the addition of the link time would help improve understanding as this may be confusing otherwise. For the coherent experiment the link time is $(1/1250) \text{ s} = 800 \mu\text{s}$. The text has been updated accordingly from: “When investigating the impact of the matter-link on the T_{2^*} time, a variable number $N=\{2,100\}$ of qubit transport operations can be undertaken within the delay time τ , such that $NT_{\{L\}} < \tau$, where $T_{\{L\}}$ is the time taken for one link.”. To read on page 15, lines 24-27, in the revised manuscript: “When investigating the impact of the matter-link on the T_{2^*} time, a variable number $N=\{2,100\}$ of qubit transport operations can be undertaken within the delay time τ , such that $NT_{\{L\}} < \tau$, where $T_{\{L\}}$ is the time taken for one link (800 μs).”

The referee states: “Page 16, line 5-7 “The axial trap frequency is $\nu_{\text{ax}} = \omega_{\text{ax}}/2\pi = 141(1)$ kHz and the radial frequencies are $\nu_{\text{rad}} = \omega_{\text{rad}}/2\pi = 1.15(3)$ and $1.31(3)$ MHz.” I feel that this information is important enough to be added to the main text. I was effectively looking for it since motional mode frequencies are strictly connected to ion transport.”

We agree with the referee that it would be informative to have the axial and radial frequencies as part of the main body of the text. The ‘Experimental Set-up’ section has been modified on page 9, lines 10-12, to include the sentence: “The axial trap frequency is $\nu_{\{\text{ax}\}} = \omega_{\{\text{ax}\}}/2\pi = 141(1)$ kHz and the radial frequencies are $\nu_{\{\text{rad}\}} = \omega_{\{\text{rad}\}}/2\pi = 1.15(3)$ and $1.31(3)$ MHz.”

The referee states: “Page 16, line 11-12 "The cost function is constructed to fulfil the following criteria:" the information on the cost function is vague, what is the relative weight for the three subcost function? Have they been simply summed? Sum of the exponentials? I think more mathematical details are required to be able to reproduce it given that the transport is at the heart of the subject of the manuscript.”

We are thankful to the referee for pointing this out. In order to address the referee’s comments, the description of the cost function has been rephrased to use terminology that is more common for constrained optimisation problems. We now state, in the revised manuscript, on page 18, lines 1-8: “The minimisation problem is constructed from a cost function, which aims to minimise the sum of the squares of the voltages across all electrodes. In addition, two constraint functions are also aimed to be minimised. These consist of: 1. the electric field at the ion position, and 2. the axial electric field curvature (and thus the axial secular frequency). Each constraint function is weighted with penalty factors $p_{\{1\}}$ and $p_{\{2\}}$ respectively such that the parameter that is to be minimised is normalised. This is due to several orders of magnitude in difference in the numerical values of each of these parameters. These penalty factors remain constant throughout the optimisation.”

Here, the relative penalties for the simulations of the waveform used in this experiment are:

- Penalty (Sum of the squares of the voltages) = 1 (cost function)
- P1 (Electric field at the ion position): $1e6$
- P2 (Curvature): $2e6$

However, these parameters depend on what curvature, and residual electric field one decides upon when simulating a shuttling sequence such that we feel that it is best for them not to be included in the revised manuscript as these may vary from one implementation to the next.

In addition, we have also added the sentence on page 17, line 15 and page 18, line 1: “...implemented using the wrapper of the algorithm in Ref. [50] in the SciPy python library” to give the reader a better understanding of how the constrained optimisation is undertaken. For this a reference to the original paper (Kraft, Dieter. "A software package for sequential quadratic programming." *Forschungsbericht- Deutsche Forschungs- und Versuchsanstalt fur Luft- und Raumfahrt* (1988)) on the SLSQP algorithm has been added as Ref. 50. Finally, the words “minimisations” and “cost function” have been replaced by the words “solution” and “minimisation problem” respectively on page 18, line 19, in the revised manuscript.

With regards to comments of the previous referee #2, the referee states: “I think the newly introduced discussion from page 6 lines 13-21 is well written, but still lacks a physical explanation for how the drifts might arise for the non-experts, i.e. why are they actually expected to happen.”

The referee is correct. In the text we mentioned that the piezo actuators can be used to compensate for drifts in the alignment of the modules, but did not elaborate on the sources of drift whose knowledge would be useful to non-experts. Here, module drift is expected to arise from temperature changes which are likely to occur as a result of thermal cycles between cryogenic and room temperature, fluctuations in the available cooling power provided by a cryogenic system, or changes in the power dissipated by an ion trap QCCD module. These temperature changes then lead to module displacements through thermo-mechanical effects, i.e. thermal contraction and expansion of the system. To make this clearer in the manuscript, the text has been amended to mention that the source of module drift is driven by temperature changes, an explanation which we believe should be sufficient to non-experts. The text now reads on page 7, line 1-3: “These may be engineered from a combination of compact shear piezo actuators, providing the sufficient travel range to compensate for module drift that can arise from temperature changes while maintaining high module alignment accuracy.”

With regards to comments of the previous referee #2, the referee states: “I think this comment has been discussed by the authors, but my own personal opinion still differs, see my comment number 1).”

We hope this has now been satisfactorily answered given the changes that we have made in response to the referee’s comment number 1).

With regards to comments of the previous referee #2, the referee states: “The new detailed data on the trap structure is indeed very useful, I think that a useful information for reproducibility that is still missing would be the thickness of the electrodes.”

We agree with the referee that this would indeed be informative to the reader from a reproducibility standpoint. Here, both RF and DC electrodes have a thickness of 1 μm . The Method section in the manuscript has been revised accordingly to now read on page 15, lines 8-9:

“Each module features a pair of RF electrodes that are 270 μm wide, 1 μm thick and are separated by 90 μm ...” and further down on lines 10-12: “Confinement in the axial direction is provided by outer segmented static voltage electrodes with width 220 μm , thickness 1 μm and separated by a gap of 10 μm .”

With regards to comments of the previous referee #2, the referee states: “I think this comment has not yet been properly addressed, see my own comment number 8).”

Given the amendments the authors have made in response to comment number 8), we hope this has now been sufficiently and adequately answered.

Summary of additional amendments that have been made to the main text:

1. Page 2, lines 10-11: “realised” has been changed to “demonstrated”.
2. Page 2, lines 15-16: “quantum charged-couple device” has been corrected to “quantum charge-coupled device”.
3. Page 4, line 9: “a 3 level distillation process” has been changed to “a 3-level distillation process”.
4. Page 7, line 35: “Each module is 17.5 x 13.5 mm in size” has been changed to “Each module is 17.5 x 13.5 mm² in size”.
5. Page 8, line 8: “369 nm of” has been amended to “369 nm laser light off”.
6. Page 8, line 28: an error on the statement “RF voltages are applied at an amplitude $V_0 = 203.5\text{V}$ ”, has been rectified to “RF voltages are applied at an amplitude $V_0 = 101.75\text{V}$ ”.
7. Page 9, lines 4-5: “for ion transport success fidelity measurements” has been changed to “to measure the infidelity associated with ion loss during transport”.

8. Page 9, lines 10-12: “thereby placing a lower limit of 99.999993% on the ion transport success fidelity” has been replaced by “thereby placing an upper limit of 7×10^{-8} on the infidelity associated with ion loss during transport”.
9. Page 9, lines 16-17: “ion-transport success fidelity” has been changed to “infidelity associated with ion loss during inter-module transport”.
10. Page 9, lines 18-19: “this fidelity” has been changed to “the probability of successful transport”.
11. Page 10, line 4: “The ion transport success fidelity” has been changed to “the infidelity associated with ion loss during inter-module transport”.
12. Page 11, line 2: “10.000(1) G” has been rectified to the correct value of 10.177(1) G”.
13. Page 12, line 10: “for the case” has been removed.
14. Page 13, line 5: the uncertainty on the $T2^*$ measurement of “560(60) ms” has been corrected to “560(40) ms”.
15. We have included an upper bound for the loss of coherence of the quantum state during ion transport on page 13 lines 10-15. This reads: “Figure 4a can be used to determine an upper bound for the loss of coherence for the qubit during transport. The lower bound of the fringe contrast for 100 links is 0.95. This translates to an upper bound on the infidelity per link of 5×10^{-4} . This confirms that the matter-link can be implemented to facilitate information transfer with a high quantum-state fidelity. The actual infidelity per link is expected to be orders of magnitude lower as shown in the experimental demonstration for linear ion transport by Kaufmann et al. [Ref. 46]”.
16. Page 16, line 61: “369.5 Doppler cooling laser” has been changed to “369.5 nm Doppler cooling laser”.
17. Page 14, line 5: “as well as control hardware” has been changed to “as well as the control hardware”.
18. Page 14, line 5: “therefore” has been removed.

19. Page 14, lines 6-7: “a similar technique” has been changed to “transport with sub-quanta motional excitation”.
20. Page 17, lines 12-13: “Finite Element Method (FEM)” has been changed to “FEM” as it had already been mentioned in the text.
21. Page 18, lines 1-3: In line with the editorial policy of Nature Communications, we have included a “DATA AVAILABILITY” section. The section reads: “The data that support the findings of the study are available from the corresponding authors upon available request.”. This data may be requested by any reader at any time.

Finally, we want to share again our sincere appreciation of the time that all the referees have set aside for the review of this manuscript. We hope that the additional amendments that we have made and the responses that we have provided to their comments answer clearly and adequately any of their outstanding concerns and we hope that our manuscript is now ready for publication.

REVIEWERS' COMMENTS

Reviewer #4 (Remarks to the Author):

I thank the Authors for their response. All my comments have been properly addressed and I consider the manuscript suitable for publication.